# Phenotypic Broad Spectrum of Bacterial Blight Disease Resistance from Thai Indigenous Upland Rice Germplasms Implies Novel Genetic Resource for Breeding Program

Atitaya Chumpol [1], Tidarat Monkham [1], Suwita Saepaisan [2], Jirawat Sanitchon [1], Shanerin Falab [2] and Sompong Chankaew [1,*]

1   Department of Agronomy, Faculty of Agriculture, Khon Kaen University, Khon Kaen 40002, Thailand
2   Department of Entomology and Plant Pathology, Faculty of Agriculture, Khon Kaen University, Khon Kaen 40002, Thailand
*   Correspondence: somchan@kku.ac.th; Tel.: +66-85124-0427

**Abstract:** Bacterial blight (BB) disease, caused by *Xanthomonas oryzae* pv. *oryzae* (*Xoo*), is one of the most devastating diseases of rice worldwide. Breeding for BB resistance has been utilized to overcome this constraint of rice production; however, limited genetic resources of BB resistance or non-desirable genetic linkage between BB disease resistance and agronomic traits have become major obstacles. Interestingly, indigenous upland rice cultivars cultivated across Thailand are considered to be novel genetic resources of BB resistance for rice cultivar improvement through breeding programs. In this study, we screened for BB disease resistance among 256 indigenous upland rice cultivars using individual inoculation of two virulent *Xoo* isolates; NY1-1 and MS1-2, under greenhouse conditions. The results showed that 19 indigenous rice cultivars demonstrated BB disease resistance abilities after inoculation. These 19 upland rice cultivars were further examined for broad-spectrum resistance (BSR) performance through five individual *Xoo* isolate inoculations, under greenhouse conditions. Moreover, a mixed five *Xoo* isolate inoculation, including NB7-8, CM3-1, CN2-1, MS1-2, and NY1-1, was conducted to assess the BSR for BB resistance of those 19 cultivars under field conditions. Simultaneously, independent plants of the 19 varieties were grown without inoculation in the field to observe the disease reactions from the natural infection caused by local *Xoo* isolates. The results of the three experiments herein showed that five indigenous upland rice cultivars—ULR024, ULR029, ULR172, ULR207, and ULR356—consistently expressed 100% BSR to BB disease, as well as the resistance check varieties, IRBB5. This further illustrated that ULR024, ULR029, ULR172, ULR207, and ULR356 upland rice cultivars were phenotypically resistant to five *Xoo* isolates, within those (a) inoculated with five individual isolates under greenhouse conditions or (b) inoculation with five mixed isolates under field conditions. Moreover, the ULR024, ULR029, ULR172, ULR207, and ULR356 revealed BB disease-resistance abilities to natural infection. These results present novel genetic resources from indigenous upland rice cultivars in further breeding program of BB resistance in varied rice cultivars.

**Keywords:** clipping method; *Xanthomonas oryzae*; selection; genetic resources; BSR

## 1. Introduction

Rice (*Oryza sativa* L.) is a staple cereal crop consumed by over half of the world's population, particularly in Asia [1]. However, one of the major adverse factors affecting rice productivity is bacterial blight (BB) disease caused by *Xanthomonas oryzae* pv. *oryzae* (*Xoo*) [2–4], as found in Southeast Asia in countries such as Indonesia, the Philippines, Thailand, and Vietnam. Over 30 pathovars of *Xoo* have been reported across global rice cultivation areas [5–7]. *Xoo* can infect rice plants at an early stage, younger than 21 days after transplanting, through plant roots and leaves under high relative humidity, resulting in seedling wilt symptom or Kresek [4]. The severe symptoms occur at the tillering stage,

notably showing tannish-gray to white lesions along the leaf veins [4,8], leading to complete crop losses in rice production [2].

Upland rice cultivars have been grown in particular areas with inadequate water availability, such as in highlands and high elevation areas, comprising roughly 20 million hectares in the intertropical regions, including about 60%, 30%, and 10% in Asia, Latin America, and Africa, respectively [9]. In Thailand, upland rice has been predominantly cultivated in the north and the northeast regions of the country, where precipitation is lower than other areas [10]. All among upland rice cultivars grown in Thailand, the Sakon Nakhon (SKN) rice cultivar was only one of the improved cultivars released by the Department of Agriculture (DOA) in 2000. This cultivar was derived from a cross between Hom Om and RD10, producing high yields with a favorable aroma. It is able to grow in both lowland and upland conditions [11]. However, the SKN rice cultivar has been susceptible to BB disease resulting in significant crop damages [12]. To date, none of BB-resistant cultivars of any upland rice cultivars have been reported or improved upon in Thailand.

The BB-resistant upland rice cultivars, carrying the major resistance (*R*) genes, are needed as genetic resources of breeding program for BB disease resistance improvement for both lowland and upland rice cultivars. Several methods of breeding have been employed to develop durable- and broad-spectrum BB disease-resistant lowland rice cultivars. For example, conventional breeding, aided by marker-assisted selection (MAS) through gene pyramiding, resulted in durable resistances of many rice varieties to different races of *Xoo* [13–18]. The International Rice Research Institute (IRRI) released several BB-resistant cultivars, such as the IRBB series (IRBB5, IRBB7, IRBB21, etc.), which carry the *xa5*, *Xa7*, and *Xa21* genes for bacterial blight resistance. They became popular donor parents in rice improvement programs for BB resistance worldwide. However, all IRBB resistance series were cultivated in irrigated lowland paddy fields. The exploration of BB-resistant upland rice cultivars is, therefore, still needed.

In addition to diversity of *Xoo*, diverse resistant rice cultivars may possess different resistance genes for BB disease, thus varying in BB resistance abilities [19,20]. More than 40 BB resistance genes have been identified across various rice cultivars [21]. While the *Xoo* races continue to evolve with various escape plant-resistant mechanisms, developed rice cultivars containing a single resistant gene would no longer be effective against evolved *Xoo* strains [4]. Indeed, breeding programs through gene pyramiding can advantage a broad spectrum for BB resistance caused by different *Xoo* strains in an improved rice cultivar [4,15–18], where a broad spectrum of disease (BB) resistance can attribute the sustainability of crop improvement [22–24]. Therefore, it is crucial to explore the new parental resistant resources to develop a broad spectrum for BB-resistant rice cultivars.

The objective of this research was to explore BB disease resistance abilities, using two virulent *Xoo* isolates, collected over 256 Thai indigenous upland rice germplasms. Furthermore, the screened upland rice germplasms demonstrating BB resistance were further examined for broad-spectrum BB resistance through multiple *Xoo* isolate inoculation (single isolate inoculation using five *Xoo* isolates on leaf of the same plant) under greenhouse conditions and a mix of five *Xoo* isolate inoculation under field conditions. This will provide a source of rice-resistant cultivars impervious to BB disease from Thai indigenous upland rice germplasm in future rice breeding programs.

## 2. Materials and Methods

### 2.1. Plant Materials

In total, 256 indigenous rice cultivars were primarily employed to screen BB resistance under greenhouse conditions. Such cultivars were collected from local growing area across Thailand (Figure 1), where BB resistance would be differentiated among their original populations. In the single isolate inoculation experiment, the BB resistance abilities of six resistant cultivars (IRBB5, IRBB21, IR62266, PRT, PLD, and SR1) and three susceptible check cultivars (KDML105, RD6, and RD23) were compared. Four resistant (IRBB5, IRBB21,

IR62266, and SR1) and six susceptible check cultivars (KDML105, ULR024, ULR089, SKN, RD6, and RD23) were examined in the multiple isolate inoculation and field experiment.

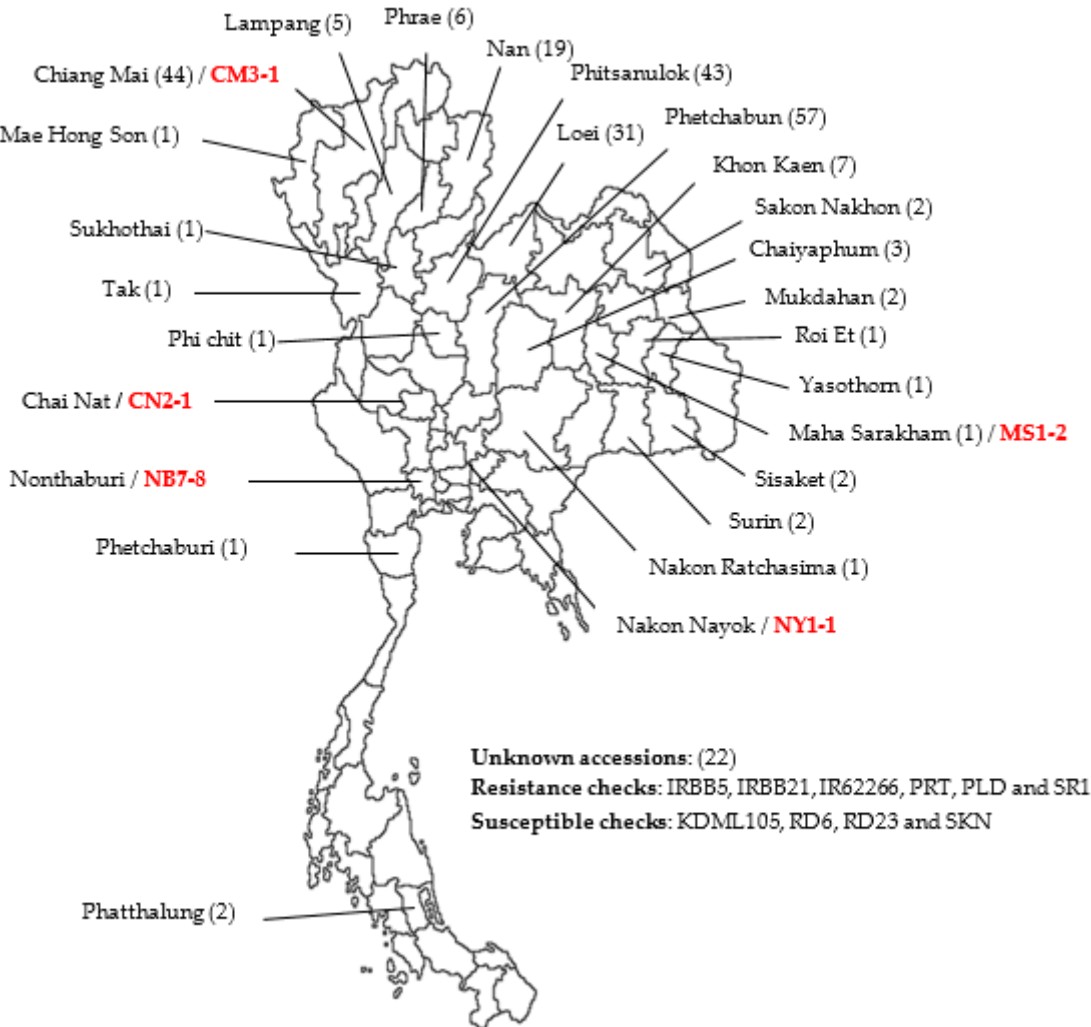

**Figure 1.** Sources of indigenous upland rice germplasms used in this study. The number of collected accessions is shown inside the bracket next to each province. The origins of the BB isolates are shown in bold red print.

*2.2. Xanthomonas oryzae pv. oryzae Isolates*

The five different originated isolates of *Xoo* used in this study, CM3-1, CN2-1, MS1-2, NB7-8, and NY1-1, were provided by Assistant Professor Dr. Sujin Patarapuwadol of Kasetsart University, Nakhon Pathom Province, Thailand (Figure 1). Each was isolated from various diseased rice cultivars and maintained in laboratory. A pure culture of every single *Xoo* isolate was multiplied on nutrient agar at 25–27 °C for 72 h before preparing the *Xoo* inoculum.

*2.3. Screening for BB Resistance of Indigenous Upland Rice Germplasms*

The 256 indigenous rice cultivars as well as the check cultivars were inoculated with two individual *Xoo* isolates (MS1-2 and NY1-1) in the greenhouse experiments. *Xoo* inoculation was conducted within 96 hill seedling trays. Seeds of the tested rice cultivars were individually sown in respective hills containing clay soil. Of the 96 hills, two rows next to the border were sown with KDML 105, which is often used as a susceptible variety for trapping the *Xoo*. The remaining 60 were randomly sown with 20 tested cultivars, each containing three hills (replications), relying on completely randomized design (CRD).

Fertilizer was applied at 14 and 20 days after planting (DAP) with 28.12 kg/ha $N_2$, $P_2O_5$, $K_2O$. Two seedlings aged 21 days after sowing (DAS), per hill, were inoculated with bacterial inoculum via the clipping method [25] (Figure 2). The *Xoo* inoculum was prepared as suspension by mixing bacterial colonies grown on nutrient agar with sterile distilled water. The final concentration of *Xoo* suspension was adjusted into $OD_{600} = 0.60$ using a spectrophotometer [26]. Inoculated plants were further grown in the greenhouse under a mist nozzle. All experiments were conducted during the rainy season from July to August 2015 and 2016 at Khon Kaen University, Khon Kaen, Thailand (Figure 2). BB resistance was evaluated through disease scoring based on lesion length on symptomatic rice leaves at Day 14 post-inoculation. The standard method of the International Rice Research Institute (IRRI) [27] was applied to categorize resistance and susceptibility groups of the tested indigenous rice cultivars. Cultivars with a mean lesion lengths of <10 cm were considered as resistant and >10 cm were considered as susceptible.

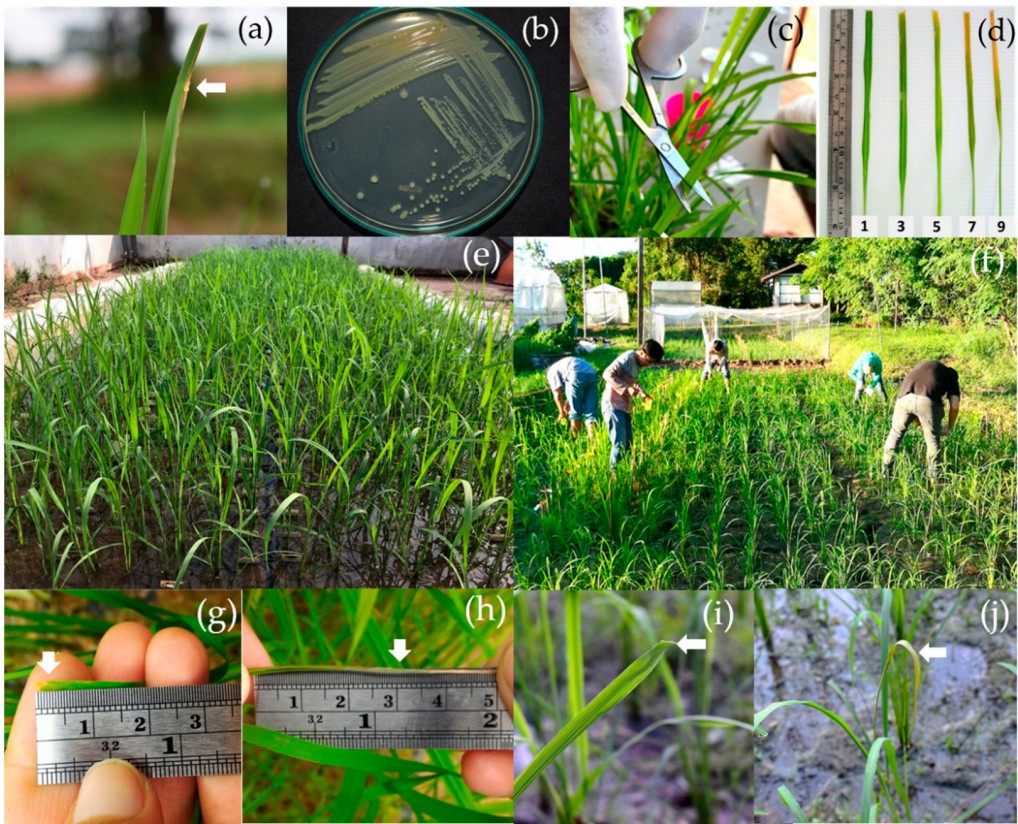

**Figure 2.** *Xoo* inoculation and evaluation of BB disease resistance of selected rice cultivars in greenhouse and field conditions at Khon Kaen University: (**a**) BB disease symptom on rice showing bacterial ooze (white arrow); (**b**) pure culture of *Xoo* on NA; (**c**) inoculation of *Xoo* by leaf clipping method; (**d**) BB-lesion scoring scales on rice leaves ranged from 1 to 9; (**e**) greenhouse experiment; (**f**) field experiment; (**g**) the BB-lesion lengths of resistant cultivar in greenhouse; (**h**), the BB-lesion lengths of susceptible cultivar in greenhouse; (**i**) the BB symptom of resistant cultivar in the field; and (**j**) the BB symptom of susceptible cultivar in field. The white arrow indicated the lesion of BB disease.

*2.4. Resistance Abilities of Upland Rice to the Five Individual Xoo Isolates under Greenhouse Conditions*

After the screening experiment, 19 rice cultivars demonstrating consistency in BB resistance were further tested for multiple *Xoo* isolate inoculation to observe broad-spectrum resistance. Here, the *Xoo* inoculum of each isolate was prepared as described in the screening experiment. The inoculum of each isolate was individually inoculated on each selected rice cultivars through clipping method following Kauffman et al. [25]. Here, the BB resistance was evaluated as described in the screening experiment.

*2.5. Resistance Abilities of Upland Rice to Five Mixed Xoo Isolates in the Field*

In the field experiment, the five mixed *Xoo* isolates were inoculated. Here, the same set of selected rice cultivars used in multiple *Xoo* isolate inoculation experiment were evaluated to confirm BB resistance ability at the agronomy field crop station at Khon Kaen University. Seedlings at 28 DAS were transplanted into a two-meter-long single-row plots surrounded by the trapping cultivars (RD6). Each row of particular rice cultivars was assigned as a replication. Three replications of randomized complete block design (RCBD) were conducted with 25 × 25 cm spacing between plants. Nitrogen fertilizer ($N_2$), at the rate of 75 kg/ha, was applied three times: at 9, 11, and 14 days after planting. Each five single isolate inoculum was prepared as described above, then gently mixed together with the same volume. The final concentration of mixed isolate inoculum was also $OD_{600}$ = 0.60, and the clipping method was also applied to inoculate bacterial suspension onto the rice plants [25] (Figure 2). In the field experiment, disease scoring (0–9) of the BB was done at the tilling stage following IRRI [27]. The cultivars were classified to resistant levels by mean BB score: 0–3 = resistant, >3–5 = moderately resistant, >5–7 = moderately susceptible, and >7–9 = susceptible. Additionally, the amount of rain, relative humidity (RH), and temperature during the crop cycles were also recorded (Figure S1).

*2.6. Resistance Ability of Upland Rice to Natural Xoo in the Field*

The other set of such selected cultivars were grown in another field to observe the natural *Xoo* infection, after which BB resistance abilities were evaluated. Here, the field experiment was conducted as described in the mixed five *Xoo* isolates experiment. Additionally, the BB resistance was evaluated as described in the mixed five *Xoo* isolates experiment.

*2.7. Data Analysis*

The scatterplot was employed to visualize the frequency of the rice cultivars showing various lesion lengths due to the NY1-1 and MS1-2 isolates of *Xoo* infection in the single isolate inoculation experiment. The difference among means of lesion lengths observed on inoculated plants from all experiments were subjected to analysis of variance via the Statistics 10$^{©}$ (1985–2013) program (Analytical Software, Tallahassee, FL, USA). Pair-wise comparisons of those means were conducted via Tukey's Honestly Significant Difference (HSD) at a 95% confidence level. Broad-spectrum resistance (BSR) was calculated using the formula according to Ahn [28]. Additionally, the coefficient of correlation (r) of field BB scores was calculated using the formula according to Gomez and Gomez [29] via the Statistics 10$^{©}$ (1985–2013) program (Analytical Software, Tallahassee, FL, USA).

## 3. Results

*3.1. Screening for BB Resistance of Indigenous Upland Rice Germplasms*

Overall, 256 indigenous rice cultivars, inoculated with either the MS1-2 or NY1-1 isolate, exhibited bacterial blight symptom with variations in lesion length. When determining the number of rice cultivars presenting intervals of lesion lengths, more than 150 cultivars showed lesions ranging between 5 and 10 cm in length. This indicated a high frequency of upland rice cultivars showing moderate resistance to both the MS1-2 and NY1-1 *Xoo* isolates (Figure 3). Interestingly, some rice cultivars exhibited resistance to BB disease, caused by MS1-2 or NY1-1 isolate, with lesion lengths less than 5 cm (Figure 2).

The scatterplot analysis displayed positive linear correlations between lesion lengths of 256 rice cultivars, with all check cultivars, in response to the NY1-1 and MS1-2 isolates (r = 0.586) (Figure 4). This further led to the corresponding BB disease reactions of all tested rice cultivars when confronted different *Xoo* isolates. Here, five upland rice germplasms (ULR206, ULR029, ULR244, ULR304, and ULR356) performed *Xoo* resistance (Figure 4). When employing the standardized lesion length less of 7 cm to the resistance check IR62266, 19 rice cultivars (ULR024, ULR029, ULR042, ULR048, ULR092, ULR119, ULR172, ULR174, ULR181, ULR183, ULR186, ULR207, ULR222, ULR244, ULR292, ULR296, ULR305, ULR337, and ULR356) were selected to further test for broad-spectrum resistance via both

multiple isolate inoculations, under greenhouse and mixed five isolate inoculation under field conditions.

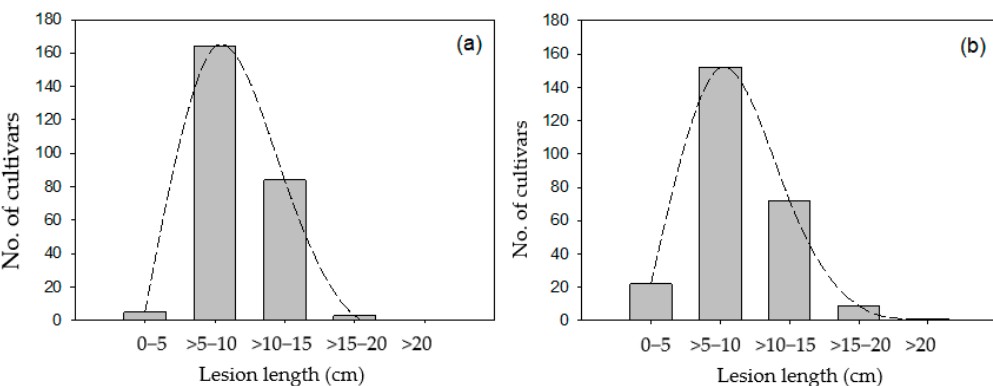

**Figure 3.** The frequencies of indigenous upland rice cultivars expressing bacterial blight disease reactions (lesion length) to two *Xoo* isolates: (**a**) MS1-2 and (**b**) NY1-1, where 0–5 cm = resistant, >5–10 cm = moderately resistant, >10–15 cm = moderately susceptible, >15–20 cm = susceptible, and >20 cm = highly susceptible.

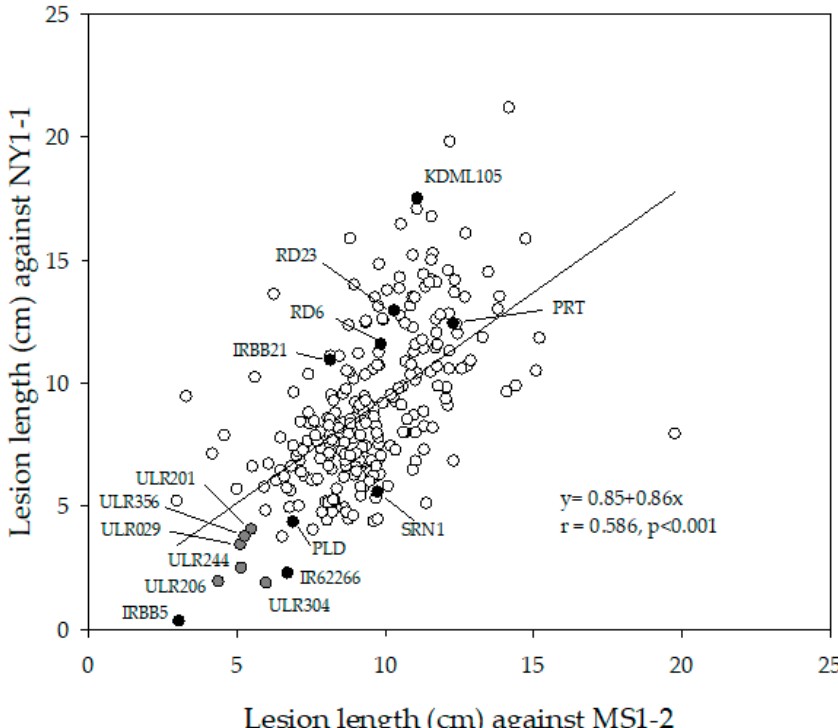

**Figure 4.** Scatterplot of lesion lengths from 256 upland rice germplasms in response to the NY1-1 and MS1-2 isolates.

*3.2. Resistance Abilities of Upland Rice to the Five Individual Xoo Isolates under Greenhouse Conditions*

All inoculated rice plants of 19 cultivars developed BB disease with variations in lesion lengths among tested rice cultivars and check cultivars for every single isolate (Table 1). The 19 tested cultivars showed lesion lengths of less than 10 cm when tested with three individual *Xoo* isolates (CM3-1, MS1-2, and NB7-8), indicating the broadly resistant reactions. However, four rice cultivars (ULR042, ULR092, ULR174, and ULR222) developed lesion lengths larger than 10 cm after testing with the isolate CN2-1, implying susceptible reactions. Only the URL092 cultivars exhibited lesion lengths larger than 10 cm after inoculation with both the isolate CN2-1 and NY1-1. When considering the BSR of

selected rice cultivars across five *Xoo* isolates, 15 of 19 rice cultivars expressed 100% BSR (Table 1). Some resistant (IRBB5) and susceptible checks (KDML105 and SKN) revealed resistance with 100% BSR and susceptibility without BSR, respectively (Table 1).

**Table 1.** The lesion lengths of 19 selected indigenous upland rice cultivars with four resistant and six susceptible check cultivars against five *Xoo* isolates tested under greenhouse conditions.

| Rice Cultivars | Mean Lesion Length (cm) | | | | | BSR (%) |
|---|---|---|---|---|---|---|
| | Isolate NB7-8 | Isolate CM3-1 | Isolate CN2-1 | Isolate MS1-2 | Isolate NY1-1 | |
| ULR024 | 4.97 def | 4.50 e–h | 5.79 efg | 5.73 e–h | 3.92 f–i | 100 |
| ULR029 | 3.58 ef | 2.05 fgh | 5.61 fg | 4.51 fgh | 8.92 c–g | 100 |
| ULR042 | 6.58 c–f | 8.00 a–f | 12.17 a–e | 6.83 d–h | 7.17 e–i | 80 |
| ULR048 | 7.28 c–f | 5.61 d–h | 9.45 b–f | 6.56 d–h | 9.33 b–g | 100 |
| ULR092 | 6.48 c–f | 7.67 a–g | 10.72 a–f | 8.79 b–g | 10.17 b–f | 60 |
| ULR119 | 4.71 def | 5.08 e–h | 6.78 d–g | 4.63 fgh | 8.11 d–h | 100 |
| ULR172 | 3.63 ef | 2.72 fgh | 7.00 d–g | 3.42 fgh | 5.89 f–i | 100 |
| ULR174 | 9.39 a–e | 7.61 a–g | 10.97 a–f | 8.09 b–h | 9.39 b–g | 80 |
| ULR181 | 1.51 f | 4.08 fgh | 4.56 fg | 5.25 fgh | 3.28 f–i | 100 |
| ULR183 | 6.42 c–f | 3.92 fgh | 6.59 d–g | 4.07 fgh | 4.48 f–i | 100 |
| ULR186 | 7.75 c–f | 5.06 e–h | 8.22 c–f | 8.67 b–g | 7.67 d–i | 100 |
| ULR207 | 2.82 ef | 1.75 fgh | 4.75 fg | 3.17 gh | 5.39 f–i | 100 |
| ULR222 | 9.97 a–e | 7.22 b–g | 10.53 a–f | 8.09 c–h | 9.39 b–g | 80 |
| ULR244 | 4.00 def | 5.78 c–h | 7.06 d–g | 6.32 d–h | 4.13 f–i | 100 |
| ULR292 | 4.42 def | 4.50 e–h | 8.00 c–f | 5.08 fgh | 7.00 e–i | 100 |
| ULR296 | 3.00 ef | 3.55 fgh | 7.00 d–g | 5.95 e–h | 6.94 e–i | 100 |
| ULR305 | 3.96 def | 3.67 fgh | 7.58 c–g | 6.11 d–h | 7.36 e–i | 100 |
| ULR337 | 6.65 c–f | 4.89 e–h | 6.83 d–g | 7.17 d–h | 5.61 f–i | 100 |
| ULR356 | 2.99 ef | 2.13 fgh | 6.36 d–g | 7.00 d–h | 7.06 e–i | 100 |
| *ULR014* | 12.78 abc | 14.22 a | 16.91 a | 13.05 a–d | 17.11 a | 0 |
| *ULR089* | 12.61 abc | 11.06 a–e | 13.89 abc | 16.89 a | 14.56 a–d | 0 |
| *RD6* | 8.98 b–e | 12.50 abc | 9.93 b–f | 12.71 a–e | 13.34 a–e | 40 |
| *SKN* | 16.15 ab | 11.89 a–d | 15.61 ab | 15.12 ab | 15.03 abc | 0 |
| *KDML105* | 16.22 a | 13.56 ab | 15.78 ab | 14.94 abc | 15.95 ab | 0 |
| *RD23* | 11.03 a–d | 4.67 e–h | 12.50 a–d | 10.33 a–f | 5.08 f–i | 40 |
| **IR62266** | 3.04 ef | 1.11 gh | 6.11 d–g | 6.58 d–h | 1.43 hi | 100 |
| **IRBB21** | 8.87 cde | 6.22 c–h | 7.70 c–g | 6.67 d–h | 6.94 e–i | 100 |
| **IRBB5** | 1.37 f | 0.42 h | 1.33 g | 1.15 h | 0.75 i | 100 |
| **SR1** | 7.95 c–f | 3.06 fgh | 12.11 a–e | 9.69 b–g | 3.00 ghi | 80 |
| F-test | ** | ** | ** | ** | ** | - |
| CV (%) | 33.53 | 36.29 | 23.07 | 28.42 | 28.29 | - |

The letters after each value represent the significant level within each row, ** significant at $p < 0.01$. Cultivars listed in italic are susceptible checks, while bold letters represent are resistant. Mean lesion lengths: <10 cm = resistant; >10 = susceptible. BSR = broad-spectrum resistance. CV = coefficient of variation.

### 3.3. Resistance Abilities of Upland Rice to Mixed Five Xoo Isolates in the Field

After individual five isolate inoculation experiment under greenhouse conditions, the 19 tested cultivars were inoculated with mixed five isolates to evaluate BB resistance ability via BB lesion scoring, according to IRRI (1996), in the field. Under this condition, only six rice cultivars (ULR024, ULR029, ULR172, ULR207, ULR337, and ULR356) demonstrated BB scores lower than five, implying the resistance ability to the mixed five *Xoo* isolates (Table 2). The nine rice cultivars (ULR042, ULR048, URL092, URL174, ULR183, ULR186, URL222, ULR292, and ULR296) performed BB susceptibility in this experiment (Table 2). Four of them (ULR042, URL092, URL174, and URL222) were consistent to the results of individual five isolate inoculation experiment under greenhouse conditions with BSRs 40–80% (Table 1). Susceptible check cultivars showed susceptibility to the five mixed isolates with high BB scores, while only one resistant check, IRBB5, revealed BB disease resistance in this experiment.

**Table 2.** Mean scores of bacterial blight resistance of 19 selected indigenous upland rice varieties with four resistant and six susceptible check varieties under filed conditions at Khon Kaen University, Khon Kaen, Thailand.

| Cultivars | Mean Score of Bacterial Blight Resistance | | | |
|---|---|---|---|---|
| | Mixed Isolate Inoculation | Reaction | Natural Infection | Reaction |
| ULR024 | 3.33 ij | R | 0.00 d | R |
| ULR029 | 4.67 d–i | MR | 1.70 d | R |
| ULR042 | 8.67 ab | S | 8.33 ab | S |
| ULR048 | 8.67 ab | S | 6.67 ab | S |
| ULR092 | 8.83 a | S | 7.33 ab | S |
| ULR119 | 5.17 c–i | MS | 0.37 d | R |
| ULR172 | 4.00 g–j | MR | 0.00 d | R |
| ULR174 | 7.67 a–d | S | 8.17 ab | S |
| ULR181 | 5.67 b–h | MS | 0.27 d | R |
| ULR183 | 7.00 a–g | S | 2.50 cd | R |
| ULR186 | 7.67 a–d | S | 5.83 abc | MS |
| ULR207 | 4.50 e–i | MR | 0.00 d | R |
| ULR222 | 8.00 abc | S | 6.07 ab | MS |
| ULR244 | 6.17 a–i | MS | 2.10 d | R |
| ULR292 | 7.50 a–e | S | 0.80 d | R |
| ULR296 | 7.17 a–f | S | 0.00 d | R |
| ULR305 | 6.67 a–h | MS | 0.27 d | R |
| ULR337 | 4.17 f–i | MR | 1.03 d | R |
| ULR356 | 3.83 hij | MR | 0.67 d | R |
| *ULR014* | 9.00 a | S | 8.33 ab | S |
| *ULR089* | 8.83 a | S | 8.00 ab | S |
| *KDML105* | 9.00 a | S | 7.83 ab | S |
| *RD23* | 9.00 a | S | 8.67 ab | S |
| *RD6* | 9.00 a | S | 8.50 ab | S |
| *SKN* | 9.00 a | S | 9.00 a | S |
| **IR62266** | 6.00 a–i | MS | 6.17 ab | MS |
| **IRBB21** | 8.83 a | S | 6.67 ab | MS |
| **IRBB5** | 1.00 j | R | 0.00 d | R |
| **SR1** | 8.67 ab | S | 5.50 bc | MS |
| F-test | ** | - | ** | - |
| CV (%) | 14.46 | - | 25.38 | - |
| r | | 0.775 ** | | |

The letters after each value were considered significant within each row, ** significant at $p < 0.01$. Rice cultivars listed in italics are susceptible checks, while bold letters represent resistance. Mean BB score: 0–3 = resistant; 3–5 = moderately resistant; 5–7 = moderately susceptible; 7–9 = susceptible, according to IRRI [27]. CV = coefficient of variation, r = coefficient of correlation.

*3.4. Resistance Ability of Upland Rice to Natural Xoo in the Field*

Another set of rice plants of the 19 tested and check cultivars were grown along with those used in mixed five isolate inoculation to observe BB symptom and score the lesions. This was to confirm BB resistance ability obtained from five mixed isolate inoculation experiment under field condition. Although the BB disease scores and the reaction of test cultivars showed slightly difference between mixed isolate and natural infection, the relationship of the BB scores between the two methods was positively correlated (r = 0.775) (Table 2). The results indicated that the BB resistance ability obtained from field conditions was consistent to those of natural infection.

**4. Discussion**

Breeding for BB disease resistance is an effective approach to improve rice cultivars to reduce the adverse effect of this disease on rice yields [4,30]. Several methods in breeding programs, such as marker-assisted selection [17,24,31,32] and gene pyramiding [16,22,33], have been employed to obtain the desirable BB-resistant traits [34]. In Thailand, *xa5*

and *Xa21* gene were introgressed into the lowland rice cultivars RD6 [17,31,32] and KDML105 [35], respectively. However, their introgression remains limited due to the genetic linkage drag characteristics [17,31,32]. Novel parental resources of BB disease resistance from various rice cultivars, especially upland rice cultivars, must be explored, contributing to the genetic resources of both lowland and upland rice breeding programs. Few studies have been carried out that identify BB-resistant cultivars via the observation of lesion lengths of resistant check cultivars among Thai indigenous lowland rice. Inoculations of individual *Xoo* isolates under greenhouse conditions showed BB resistance of the phenotype of the Kan Phu Daeng, Phuyai Li and RD23 lowland rice cultivars [36], and the LG6822, PRT, HMLN and PLD lowland rice cultivars [37]. In this study, the screening of BB resistance in the single *Xoo* strain inoculation experiment under greenhouse conditions, with the MS1-2 or NY1-1 isolate, revealed the BB resistance in more than 150 cultivars among 256 tested indigenous upland rice cultivars (Figure 3). Moreover, there was the positive correlation between the lesion lengths of individual cultivars in response to the MS1-2 and NY1-1 isolate (Figure 4), demonstrating the correspondence of host plants' reactions to two different virulent *Xoo* isolates as investigated on indigenous lowland rice cultivars by Sombunjitt et al. [36] and Sribunrueang et al. [26]. This presumably suggests whether the individual host plants possess the analogous resistance genes [37,38] or that the two examined *Xoo* isolates own the analogous avirulence genes [39], resulting in greater similarities in the physiological and morphological responses [31,40].

Plant disease resistance based on the gene-for-gene concept appears to be non-durable, due to the plant–pathogen arms race [41,42]. In case of BB disease in rice, developed rice cultivars containing a single resistance gene compatible with a specific *Xoo* race may no longer be effective against an evolved *Xoo* [43,44], especially for hypersensitive types of the major resistance genes [45]. In this study, the resistant check cultivars, IRBB21, IR62266, and SR1, appeared to be susceptible to the mixed isolate inoculation and natural infection experiments. This indicated that such three cultivars may express resistance breaking when faced particular *Xoo* isolates, or mixed *Xoo* isolates as reported by Kwanwah et al. [12]; Sontornkarun et al. [46] and Mishra et al. [47]. On the other hand, a single major resistance gene of specific rice cultivar might cope with diverse *Xoo* isolates such as *xa5*, making it is essentially needed as genetic resources for breeding program [31,48,49]. Evidence was presented, for instance, when the introgressed FF329 rice line obtaining *Xa39* performed the resistance ability against 21 *Xoo* isolates [45], and when the G252 rice introgression line containing *Xa47*(*t*) revealed resistance ability against 10 *Xoo* isolates [50]. In this study, 15 of the 19 selected indigenous rice cultivars, screened with MS1-2 and NY1-1 isolate inoculations, performed 100% BSR, as determined by five individual *Xoo* isolates' (CM3-1, CN2-1, MS1-2, NB7-8, NY1-1) inoculation under greenhouse conditions (Table 1). This indicated that such cultivars might possess an analogous resistance gene encountering defense mechanism against each single *Xoo* isolate [21,39]. The 100% BSR of 15 rice cultivars found in this study (Table 1) suggests the operation of multiple major gene resistance, as found in the cultivar IRBB52, IRBB53, IRBB58, and IRBB63 that possessed the resistance genes *Xa4* + *Xa21*, *xa5* + *xa13*, *xa5* + *Xa13* + *Xa21*, and *xa5* + *Xa7* + *xa13*, respectively [42].

Co-infection of two or more isolates of *Xoo* on rice plants occurs naturally [4] in agriculture fields [6]. This implies that genetically different pathogen isolates infect plants from within. Therefore, plants with multiple major gene resistance might contribute to broad-spectrum resistance of BB disease, caused by different *Xoo* isolates, as found in the Zhachanglong rice cultivar, which harbors *Xa3*/*Xa26*, *Xa22*(t), and *Xa31*(t) resistance genes, which confer resistance to multiple Chinese *Xoo* strains [51–53]. In this study, we tested this hypothesis by inoculating five mixed isolates of *Xoo* onto the 19 rice varieties tested under field conditions. Our results revealed that only 6 of the 19 selected indigenous upland rice cultivars showed resistance reaction after the *Xoo* isolates' inoculation under field conditions (Table 2). This contrasted with the results of the single isolate inoculation experiment mentioned above (Table 1). The mixed isolate inoculum of *Xoo* might affect the plant resistance through the synergistic effect of different isolates of *Xoo* [54], resulting in

varied disease symptoms and resistance reactions [55–57]. In this experiment, we excluded the effect of mixed isolate inoculum under field conditions by establishing 19 tested varieties as trap plants without inoculation. The results indicated a positive correlation between the resistance scores from mixed inoculation and natural infection (Table 2). In addition to the plant–pathogen interaction, environmental factors, such as relative humidity [26,30,58] and temperature [59,60], play a part in BB disease incidence and epidemiology, referred to as the disease triangle (Figure S1), [61]. These results follow the influence of environmental factors on optimal conditions for BB disease incidence reported by Sribunrueang et al. [26].

The BSR to BB disease of rice cultivars resulting from this study could provide the genetic resources to breeding program necessary to develop more durable resistance and long-term sustainability of resistant rice cultivars. This highlights the importance of further investigation to identify major resistance genes involved in resistance reaction from Thailand's upland rice germplasms showing 100% BSR.

## 5. Conclusions

Thai indigenous upland rice cultivars in this study varied in phenotypic BB disease responses after inoculation under both greenhouse and field conditions. Five upland rice cultivars, ULR024, ULR029, ULR172, ULR207, ULR337, and ULR356, were identified as the BSR for BB resistance in Thailand, which represent potential alternative genetic sources of BB resistance in future Thai rice breeding programs. Further research of interest includes the identification of the resistance genes evolving in BB resistance, which cultivars showing 100% BSR.

**Supplementary Materials:** The following supporting information can be downloaded at: https://www.mdpi.com/article/10.3390/agronomy12081930/s1, Figure S1: Relative humidity (RH%) (a); minimum, maximum, and average temperatures (b); and amount of rainfall (c) during field experiment at Khon Kaen University, Thailand.

**Author Contributions:** Conceptualization, A.C. and S.C.; methodology, A.C. and S.S.; software, A.C. and T.M.; validation, T.M., S.C., S.S. and J.S.; formal analysis, A.C.; investigation, S.C.; resources, S.C., T.M. and J.S.; data curation, A.C.; writing—original draft preparation, S.C. and S.F.; writing—review and editing, S.C. and S.F.; visualization, S.F. and T.M.; supervision, S.C.; project administration, S.C.; funding acquisition, S.C. All authors have read and agreed to the published version of the manuscript.

**Funding:** This research received no external funding.

**Institutional Review Board Statement:** Not applicable.

**Informed Consent Statement:** Not applicable.

**Data Availability Statement:** The data presented in this study are available on request from the corresponding author.

**Acknowledgments:** This research was financial supported by the Young Researcher Development Project of Khon Kaen University, S. Chankaew. The authors are grateful to the Plant Breeding Research Center for Sustainable Agriculture, Khon Kaen University, Khon Kaen, Thailand. Our gratitude is also extended to Assist. Sujin Patarapuwadol of Kasetsart University for providing various sources of *Xoo* isolates.

**Conflicts of Interest:** The authors declare that no conflict of interest exists.

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
