# Peer review of "Phenotypic Broad Spectrum of Bacterial Blight Disease Resistance from Thai Indigenous Upland Rice Germplasms Implies Novel Genetic Resource for Breeding Program"

_agronomy, doi:10.3390/agronomy12081930_

Round 1

Reviewer 1 Report

The article "Phenotypic Broad-Spectrum of Bacterial Blight Disease Resistance from Thai Indigenous Upland Rice Germplasms implies novel genetic resource for the breeding program" by Chumpol et al. described a study where the authors screened 256 indigenous upland rice cultivars for Bacterial blight (BB) disease, and broad-spectrum resistance against several Xoo isolates. Based on the results, the authors speculate that five varieties could be used for breeding.  Although the study presents some interesting data, it is not convincing enough to determine the resistant rice varieties. The authors do not show any figures for lesion diameters/length on leaves (qualitative data) and lesion numbers on leaves of different cultivars after Xoo inoculations.

Author Response

Dear reviewer,

Thank you for your comment and suggestion.

This study, five upland rice cultivars; ULR024, ULR029, ULR172, ULR207, ULR337, and ULR356, were identified as the BSR for BB resistance in Thailand, which represent potential, alternative genetic sources of BB resistance in future Thai rice breeding programs. Further research of interesting includes the identification of the resistance genes evolving in BB resistance, from which cultivars showing 100% BSR.

The comment of figure of five resistant cultivars, we did not taken due to the take picture is difficult and might damage the inoculated leaf. So we provided the figure of representative of overall experiment as Figure 2 in manuscript. 

Best regards

Sompong Chankaew

Reviewer 2 Report

The manuscript entitled “Phenotypic Broad-Spectrum of Bacterial Blight Disease Resistance from Thai Indigenous Upland Rice Germplasms implies novel genetic resource for breeding program” by Chumpol et al., reports screening of bacterial blight resistant upland rice germplasms based on the phenotypic interaction between 256 Thai indigenous upland rice cultivars and 5 Thai indigenous strains of Xanthomonas oryzae pv. oryzae (Xoo). The screening experiments were carried out in both greenhouse and field conditions. Five upland rice cultivars (ULR024, ULR029, ULR172, ULR207, ULR337 and ULR356) were found resistant to all the 5 Xoo strains used. I found the results clearly presented and the conclution is convincing. However, I have following concerns or suggestions.

1. How many Xoo strains have been isolated in Thailand? Is the rice resistance to the 5 used Xoo strains Broad-Spectrum?

2. The Standard Evaluation System for Rice (SES) of IRRI has been updated but the authors cited the old versions.

3. Fig 2 should be removed since it has not been used to support the results or discussion.

4. Since the xa5 gene had been introgressed into the variety RD6 (as stated on lines 255-256), how can RD6 be susceptible to the 5 Xoo strains used (Table 2, comparied with IRBB5)?

5. It would be better for the authors to use PCR to check the known genes (such as xa5, Xa21---) in the resistant cultivars (ULR024, ULR029, ULR172, ULR207, ULR337 and ULR356). This data would provide much more insights for the screened resistance and their future application in breeding programs.

6. The English writing of this MS should be improved before acceptance for publication.

Author Response

Dear reviewer,

Thank you for your comment and suggestion. 

The reviewer comments and suggestions had been response as attached files.

Best regards

Sompong Chankaew

Reviewer 3 Report

The manuscript “ Phenotypic Broad-Spectrum of Bacterial Blight Disease Re sistance from Thai Indigenous Upland Rice Germplasms implies novel genetic resource for breeding program” presented the results of the inoculation of Xanthomonas oryzae pv. oryzae   in indigenous rice cultivars from Thailand aiming to select resistance sources for breeding programs.

Although the results are interesting, there are some concerns about the methods and the presentation of the manuscript. Therefore, a revision is necessary in the following points:

1.      Materials and methods – Please improve the description of this section. As a suggestion you should put the similar topics that was presented in the results section.

Please put the detailed disease score (resistance classification) in the line 156.

The statistical methods need to be described more detailed. What statistical method did you use? (In the table is described F-test, but not here....) How did you calculate the Coefficient of correlation?

2.      Results – Tables

It is hard to read the statistics in the tables. It appears you chose a multiple mean comparison test less conservative to reject the equality of means. However, I suggest you use another test more conservative, such as HSD or Duncan. I also suggest you to include another table showing the values of ANOVA (or other) statistics.

3- Discussion

Why were the resistant check cultivars (with the exception of  IRBB5) susceptible to Xoo? Resistance breaking? Was it previously reported in other work? Please explain in the discussion section.

4- English grammar – Please it is necessary an extensive revision of english grammar and style in the manuscript.

Other points:

Line 26 – Please correct “to observe”

Line 197 – Please correct “field”

Author Response

(The authors gave the same response as above.)

Round 2

Reviewer 1 Report

I am satisfied with the author's response. I have no further comments.

Reviewer 2 Report

All my questions have been addressed.

Reviewer 3 Report

The authors presented a revised version of the manuscript. The quality of the manuscript was improved and the authors adressed the critical points.